# A Dual-Color Tyr-FISH Method for Visualizing Genes/Markers on Plant Chromosomes to Create Integrated Genetic and Cytogenetic Maps

**DOI:** 10.3390/ijms22115860

**Published:** 2021-05-30

**Authors:** Natalya Kudryavtseva, Aleksey Ermolaev, Gennady Karlov, Ilya Kirov, Masayoshi Shigyo, Shusei Sato, Ludmila Khrustaleva

**Affiliations:** 1Laboratory of Plant Cell Engineering, All-Russian Research Institute of Agricultural Biotechnology, Timiryazevskay 42 Str., 127550 Moscow, Russia; natalia.kudryavtseva92@gmail.com; 2Center of Molecular Biotechnology, Russian State Agrarian University-Moscow Timiryazev Agricultural Academy, 127550 Moscow, Russia; ermol-2012@yandex.ru; 3Laboratory of Applied Genomics and Crop Breeding, All-Russian Research Institute of Agricultural Biotechnology, 127550 Moscow, Russia; karlovg@gmail.com; 4Laboratory of Marker-Assisted and Genomic Selection of Plants, All-Russian Research Institute of Agricultural Biotechnology, 127550 Moscow, Russia; kirovez@gmail.com; 5Kurchatov Genomics Center of ARRIAB, All-Russian Research Institute of Agricultural Biotechnology, 127550 Moscow, Russia; 6Graduate School of Sciences and Technology for Innovation, Yamaguchi University, Yamaguchi 753-8515, Japan; shigyo@yamaguchi-u.ac.jp; 7Graduate School of Life Science, Tohoku University, Miyagi 980-8577, Japan; shuseis@m.tohoku.ac.jp; 8Department of Botany, Breeding and Seed Production of Garden Plants, Russian State Agrarian University-Moscow Timiryazev Agricultural Academy, Timiryazevskay 49 Str., 127550 Moscow, Russia

**Keywords:** Tyr-FISH, plant chromosome preparation, recombinant and cytogenetic maps, transcript-based markers, genome assembly, *Allium cepa*

## Abstract

In situ imaging of molecular markers on a physical chromosome is an indispensable tool for refining genetic maps and validation genome assembly at the chromosomal level. Despite the tremendous progress in genome sequencing, the plant genome assembly at the chromosome level remains a challenge. Recently developed optical and Hi-C mapping are aimed at assistance in genome assembly. For high confidence in the genome assembly at chromosome level, more independent approaches are required. The present study is aimed at refining an ultrasensitive Tyr-FISH technique and developing a reliable and simple method of in situ mapping of a short unique DNA sequences on plant chromosomes. We have carefully analyzed the critical steps of the Tyr-FISH to find out the reasons behind the flaws of this technique. The accurate visualization of markers/genes appeared to be significantly dependent on the means of chromosome slide preparation, probe design and labeling, and high stringency washing. Appropriate adjustment of these steps allowed us to detect a short DNA sequence of 1.6 Kb with a frequency of 51.6%. Based on our results, we developed a more reliable and simple protocol for dual-color Tyr-FISH visualization of unique short DNA sequences on plant chromosomes. This new protocol can allow for more accurate determination of the physical distance between markers and can be applied for faster integration of genetic and cytogenetic maps.

## 1. Introduction

Genetic linkage maps that have been used for over 100 years aided greatly *de novo* assembly of plant genomes [1,2,3,4]. Linkage maps based on recombination rate between markers can anchor de novo sequences and order small fragments into chromosome-scale sequences. However, linkage maps, which are most accurate in regions of the genome with high rates of recombination, can result in mistakes in arranging scaffolds in the region of suppressed recombination e.g., pericentromeric, knob, and subtelomeric heterochromatin [5,6,7]. The cytogenetic map can compensate for the discrepancy between the real position of the DNA sequence and its position on the genetic linkage map. Moreover, the cytogenetic map is useful in synteny and collinearity comparison between relative genomes [8,9], especially for complex-genome organisms with large amounts of repetitive DNA, such as wheat [10] or onion [11]. Third generation sequencing technologies, for instance PacBio and Oxford Nanopore, offer vast improvements over Sanger and Illumina sequencing for long reads. Now whole genomes can be quickly sequenced, which is not the case with genome assembly as this process goes slower due to its computational complexity and labor intensity. Whole-genome sequencing is much more informative when linked and oriented to chromosomes than unlinked and disordered scaffolds. Hi-C and optical mapping methods aim to assist in genome assembly at the chromosome level. Hi-C is a sequencing-based assay originally designed to inquire the 3D structure by measuring the contact frequency between all pairs of loci in the genome [12,13]. Nevertheless, the Hi-C assembly data contains a significant number of contig orientation errors when the contig is in the correct position but not in the correct orientation [14]. Optical mapping has been used for arranging the scaffolds of pseudo-chromosomes in de novo assembly of plant genomes [15]. Rearrangements such as translocations involving the ends of pseudomolecules in *Spirodela polyrhiza* genome assembly were more easily detected by the optical mapping, while inverted orientation of a sequence within a scaffold was possible to detect only via a cytogenetics approach [16].

It is obvious that the creation a reference genome at the chromosome level of resolution will probably require more independent approaches for high confidence in the genome assembly at chromosome level. In situ imaging of molecular markers on a physical chromosome for integration of chromosomal and genetic maps is an indispensable tool for refining genetic maps and validating genome assembly at the chromosomal level. BAC-FISH mapping was successfully used for genome assembly in plant species with small genomes such as tomato [6,17], grass species *Brachypodium distachyon* [18], and duckweed *Spirodela intermedia* [19]. Though BAC-FISH mapping is less applicable for large-genome species that contain a huge amount of dispersed, repetitive DNA sequences. Integration of genetic and cytogenetic maps is a necessary step for the assembly of large genomes possessing a considerable number of repetitive sequences, like *A. cepa* (1 C = 16.3 Gb of which 98% is repetitive DNA). The possible solution to this problem can be the direct in situ visualization of molecular markers that are used to guide genome assembly. However, here we encountered an obstacle in the face of the sensitivity of the FISH method. The sensitivity threshold of common FISH is 10 Kb, while the size of markers is much smaller and they often form several hundred base pairs. Tyramide-(Tyr)-FISH is an approach to increase the sensitivity of FISH [20] and which has been applied to plant chromosomes [11,21,22,23,24,25]. Yet, the method has not become routine because of indistinct in situ hybridization signals for some probes, low detection frequency, and low reproducibility among different laboratories [26]. Taking all together, development of a reproducible method for visualizing relatively short target DNA sequences on a physical chromosome to create integrated genetic and cytogenetic maps is an opportune task in the time of rapid progress in sequencing of numerous plant genomes for validation of genome assembly at the chromosomal level. In this paper we describe a reliable Tyr-FISH method for two-color detection of short DNA sequences (1.6–3.2 Kb) on compact plant chromosomes using a detailed study of all steps of the protocol, including troubleshooting analysis.

## 2. Results

### 2.1. Probe Preparation and Labeling: Reducing Background and Increasing Signal-to-Noise Ratio

The method of Tyr-FISH signal amplification based on the combination of the advantages of enzymatic deposition of many tyramide molecules and sensitive fluorescence detection requires a special approach in the creation of DNA probes. The probe should (i) be extended for specific target sequences — design of an exon-intron sequences, (ii) lack repetitive DNA, and (iii) be the proper size after labeling to easily access the target sequences.

Increasing the amount of data in different databases makes it possible to design exon-intron probes without additional sequencing of target genes via primer construction and producing genomic amplicon. Advances in genome and transcriptome sequencing and assembly provide an opportunity to detect introns and exons boundaries and positions in the target gene. The presence of unique intron sequences within a transcript-based probe having conserved exon sequences increases both the specificity and the length of the target chromosomal sequences. For the design of the probe, we selected two markers from the transcriptome map assigned to chromosome 2, which were mapped at a distance on recombinant map [27]. To design an exon-intron probe we blasted Unigene572 and Unigene5305 marker sequences against TSA (Transcriptome Shotgun Assembly) databases of *Allium cepa* in order to find the full transcript sequences. To identify exon-exon junctions in transcript and exon-intron boundaries in the genome sequence the draft genome assembly of *A. cepa* was used (S. Sato and M. Shigyo, unpublished). The repetitive DNA sequences within the designed probe DNA and subsequently cloned and sequenced genomic DNA amplicons can be located precisely in draft and complete genomic sequence contigs by computational means (Figure 1). To detect fragments of probe that have identity to repeats and transposable elements (TE) target sequence was annotated using CENSOR [28]. Information about coordinates and lengths of introns and fragments of repeats and TEs was used to design primers and obtain an exon-intron PCR-product (Table 1). The exon-intron PCR-products were cloned and sequenced (Appendix A).

Nick translation is the most popular method to create a labeled probe for ISH so far [29]. The result of Nick translation is a set of fragments of different lengths. The length distribution is an important factor for decreasing the background and increasing the probe access to target DNA. The normal range of Nick translation fragments is from 200 bp to 500 bp. Probes above this range usually result with high background and noisy signals due to non-specific sticking to the slide surface. Smaller probes would be hybridized at chromosomes non-specifically which may end up with a spotty background. Figure 2 shows the results of the Nick translation of plasmids containing genomic amplicons of Unigene572 and Unigene5305 labeled with both biotin and digoxigenin after 90 min of reaction. All probes demonstrated a full fragmentation and proper fragment size distribution (Figure 2).

### 2.2. Chromosome Preparation Is a Key Step in Successful Tyr-FISH

Our many years of experience with the Tyr-FISH method have shown that one of the key steps in obtaining a reliable fluorescent signal is proper chromosome preparation. Using Tyr-FISH, we examined how the method of preparing chromosome slides affects the frequency and sensitivity of signal detection. We carried out a comparative analysis of two methods of plant chromosome preparation: (1) squashing and (2) dropping.

In the squashing method, chromosomes are spread by squashing the plant material in 45% acetic acid between the slide and coverslip, freezing the material to the slide in liquid nitrogen, and then removing the coverslip (see Materials and Methods). For the dropping method, we used the “SteamDrop” technique which is the product of our early development [30]. It is based on obtaining a cell suspension from meristematic tissue, dropping it on the slide and using steam for better chromosome spreading. We minimized the number of washing steps to reduce the chromosome damage and cell loss. For dropping we used cell suspension in 96% ethanol. Then we added ethanol-acetic acid fixative directly to the slide. In this case, the ratio of ethanol and acetic acid in the fixative was selected depending on the chromosome spreading and the amount of the cytoplasm in the first preparation from a sample of cell suspension. This allows for customization of the degree of chromosome spreading, the amount of cytoplasm and its transparency around the chromosome. Furthermore, the long-term storage of the cell suspension in 96% ethanol did not affect the quality of the chromosome preparations.

We have implemented differential interference contrast (DIC) microscopy for the analysis of chromosome preparations. The technique was developed by Georges Nomarski in 1952 [31]. Based on the principle of interference the DIC microscopy enables us to determine the optical density of the studied object and, thus, recognize the details inaccessible to the naked eye. DIC creates a volumetric relief image corresponding to the change in the optical density of the sample and accentuating the lines and borders.

DIC microscopy revealed differences in the structure of chromosomes depending on the method of slide preparation. The dropped chromosome slides showed more pronounced relief structure with clear edge contours of chromosomes (Figure 3a) while the squashed chromosome slides were flatter and denser (Figure 3b). We can assume that the target DNA of the chromosomes prepared by the “SteamDrop” method is more preserved than the chromosomes prepared by the squashed method.

### 2.3. Quenching Endogenous Peroxidase and Horseradish Peroxidase (HRP)

The Tyr-FISH method is based on the use of the enzyme HRP which catalyzes the deposition of many molecules of a fluorophore-labeled tyramide adjacent to the immobilized HRP (Figure 4).

The principle of labeled tyramide deposition is via free radical formation, and reaction with electron-rich moieties such as tyrosine, tryptophane, etc. [32]. HRP, in the presence of low concentrations of H_2_O_2_, converts tyramides into an oxidized form with highly reactive free radicals [33]. These activated tyramides then covalently bind to tyrosine or other nuclear and cytoplasmic protein residues on the slide in close proximity to the HRP, thus, depositing many of the labeled tyramides (Figure 5).

We used two probes, namely, Unigene572 and Unigene5305, which are genomic amplicon clones labeled by haptens: biotin and digoxigenin. Haptens are small, relatively inert organic molecules that can be attached to DNA without disrupting the DNA’s hybridization properties. The probes, Unigene572 and Unigene5305, were added together into one hybridization mixture and left overnight to hybridize with the complementary DNA sequences of chromosomes. In the FISH, the probe detection is carried out simultaneously with fluorescently labeled reporter molecules such as of avidin (streptavidin) for biotin and anti-digoxigenin antibody. In contrast to the FISH method, in Tyr-FISH it is impossible to conduct simultaneous detection of two probes, since the HRP enzyme was involved in both cases: streptavidin-HRP and anti-digoxihenin-HRP. In the Tyr-FISH method, the detection of probes is carried out sequentially. Once HRP is introduced, the fluorophore-tyramide is added to identify the first probe, for instance, the Biotin labeled probe was detected by Streptavidin-HRP and visualized by deposition of tyramide-Cy3. Quenching HRP of the first layer of detection must be performed prior to detection of the next probe: anti-DIG-HRP followed by tyramide-FITC deposition.

For quenching HRP we used hydrogen peroxide (H_2_O_2_). In the active catalytic site of HRP, namely the ferric heme domain, heterologous cleavage of H_2_O_2_ occurs [34]. However, with a large excess of H_2_O_2_, iron in the HRP domain is oxidized to the state of ferrous iron, as a result of which the activity of the enzyme is quenched [35]. In order to estimate the activity of H_2_O_2_ in quenching HRP, we used 0.3% and 3% H_2_O_2_ in PBS on “SteamDrop” slides. At a lower concentration (0.3% for 30 min), enzyme activity was not inhibited because a hybridization site for the biotin-labeled Unigene572 probe was found along with digoxigenin-labeled Unigene5305 after the second round of detection with anti-DIG-HRP + tyramide-FITC on the filter FITC (Figure 6c). Slide treatment with 3% H_2_O_2_ for the same incubation time of 30 min resulted in complete quenching of peroxidase activity (Figure 7).

Hypersensitivity of Tyr-FISH requires effective suppression of unwanted background associated with the presence of endogenous peroxidase. Peroxidases are ubiquitous in nature being found in bacteria, fungi, algae, plants, and animals. The plant peroxidases, belonging to Class III peroxidase, are involved in different vital processes in a plant [35]. In order to quench endogenous peroxidase, we used 0.3% H_2_O_2_ for 30 min. Then slides were pretreated with 4% paraformaldehyde for 10 min and dehydrated with 70, 90, and 100% ethanol. After washing in TNT buffer, the tyramides were added to the slides for 15 min. For testing the presence of endogenous peroxidase activity, the chromosome preparations were incubated with both tyramide-FITC and tyramide-Cy3 at a dilution of 1:50. The chromosome preparations were stained with DAPI and analyzed with epifluorescence microscope AxioImager M2, Zeiss. Microscopic analysis of chromosomal preparations treated with 0.3% H_2_O_2_ and those without H_2_O_2_ slide treatment (control variant) did not reveal fluorescent signals in both variants. Based on the results of this experiment, the quenching of endogenous peroxidase was removed from the protocol. Due to the fact that the presence of endogenous peroxidase in other plant species is not excluded we recommend such a test for endogenous peroxidase to be carried out ahead of the work.

### 2.4. A Dual-Color Tyr-FISH Visualization of Markers

We performed the dual-color sequential Tyr-FISH on “SteamDrop” and squashed chromosome preparations. The Unigene572 genomic amplicon (3.2 Kb) was labeled with biotin-16-dUTP and detected with tyramide-Cy3 (red fluorescence) and Unigene5305 genomic amplicon (1.6 Kb) was labeled with digoxigenin-11-dUTP and detected with tyramide-FITC (green fluorescence).

The dual-color Tyr-FISH revealed twin signals arising from two sister chromatids on the long arm of chromosome 2 probing with Unigene5305 and on the short arm of the chromosome 2 probing with Unigene572 (Figure 7). On “SteamDrop” slide preparations the frequency detection of Unigene572 was 77.4%, and Unigene5305 was 51.6% (Table 2). In contrast to the results obtained on the “SteamDrop” slides, the detection frequency of Unigene572 was 35.3% and Unigene5305 was 14.7% on squashed chromosome preparations. Moreover, the frequency of signal detection on the chromosomes of both homologs within the same metaphase on “SteamDrop” preparations was 41.9% and 12.9% for Unigene572 and Unigene5305, accordingly. Meanwhile, in squashed preparations, the signal on both homologous chromosomes was detected only in 2.9% of metaphases for Unigene572, whereas no such metaphases were found for Unigene5305.

We have also analyzed the frequency of detecting signals from two probes together per metaphase (Figure 8). In “SteamDrop” preparations, signals arising from two probes on the chromosomes of both homologs within the same metaphase were detected in 12.9% of metaphases and in the case of squashed chromosome preparations, such metaphases were not found. Metaphases with signals from both probes on one homologous chromosome accounted for 36.1% (combined variants 1 and 2, Figure 8) for “SteamDrop” preparations and 8.8% for squashed preparations.

Altogether, dual-color Tyr-FISH allowed a reliable visualization of two amplicons of unique genes with high detection frequency on dropped chromosome slides. Visualization of two probes on the same chromosome was on almost every second analyzed metaphase (49.1%, combined variants 1, 2, and 3 for “SteamDrop”, Figure 8). The frequency of signal detection depended on the method of preparing chromosome slides.

### 2.5. The Integration of Recombination and Cytogenetic Maps

We aligned the chromosomal positions of Unigene 572 and Unigene 5305 with their position on the genetic map [27]. Using the DRAWID program [36] the position of the Tyr-FISH signals arising from Unigene 572 on the short arm and Unigene 5305 on the long arm were measured. Only non-overlapping chromosomes 2 were included in measurement. The relative positions of hybridization sites on chromosome arm (RPHC) were calculated in the form of the ratio of the distance between the site of hybridization and the centromere to the length of the chromosome arm. The position of signals on “SteamDrop” and squashed chromosome preparations was compared. The RPHC of Unigene572 was 60.63 ± 2.55 and 57.61 ± 3.19 on “SteamDrop” chromosomes and squashed chromosomes, respectively. The RPHC of Unigene 5305 was 52.32 ± 1.46 and 50.92 ± 5.57 on “SteamDrop” chromosomes and squashed chromosomes, respectively. Thus, the relative position of the markers on the chromosomes remains unchanged in the preparations obtained by both methods. Our observation agrees with results obtained by Wang et al. [37] on maize chromosomes. The authors reported that a preparation of squashed chromosomes produced longer chromosomes than those observed in 3D fixed cells, but the relative position of genes on chromosomes obtained by different methods remains unchanged.

The genetic positions of the markers were anchored with their position on the chromosome 2. The physical position of the markers was expressed as a percentage of fractional length (FL) from the end of the short arm of chromosome 2 and corresponded with their position in the genetic map (Figure 9).

## 3. Discussion

The dual-color Tyr-FISH visualization of two short DNA sequences on a single individual chromosome has a number of advantages: (1) more accurate determination of the physical distance between markers due to the simultaneous detection of two markers on the same chromosome; (2) short procedure of mapping; (3) high detection frequency; (4) coverage of regions with suppression of recombination; (5) the mapped sequences are located at the site of major chromosomal landmarks, e.g., centromeres, telomeres, heterochromatin.

### 3.1. Method of Chromosome Preparation Strongly Influences the Signal Detection with Tyramide-Fluorophore

In aspiration to figure out the cause of Tyr-FISH method’s sensitivity being significantly reduced by the squashed method, compared to “SteamDrop”, we examined the key difference between these two procedures. In the squashed method, cells are squashed in a drop of an aqueous solution of 45% acetic acid, whereas in the “SteamDrop” method, cells are dropped onto a slide in 96% ethanol and scattered in a fixative containing ethanol and acetic acid. Considering that the deposition of tyramides occurs via free radical formation and reaction with electron-rich moieties of protein residues of the nucleus and cytoplasm, we may suggest that the ability of these electron-rich moieties to form covalent bonds with tyramides is notably higher in “SteamDrop” than in the squashed method. The side chains of electron-rich aromatic amino acids are hydrophobic and, hence, behave differently in water and ethanol solutions: in water solutions, hydrophobic amino acids are hidden inside the protein, while the hydrophobic conditions of ethanol solution trigger proteins to unfold [38] and transit from compact to extended state. As a result, hydrophobic amino acids forming the “core” of protein become more surfaced [39]. Slide pretreatment with paraformaldehyde, which causes covalent crosslinks between molecules, and effectively gluing them together into the insoluble meshwork [40], preserves this structure of proteins for subsequent Tyr-FISH detection. Hydrophobic conditions created by ethanol in the “SteamDrop” method make side chains of electron-rich amino acids in proteins more easily accessible for the deposition of tyramides, compared to the squashed method. This can account for the difference in detection frequency of Tyr-FISH probe on slides obtained by dropped and squashed methods. Moreover, the higher availability of labeled probe DNA for annealing with short target DNA on dropped chromosomes, compared to squashed chromosomes, also contributes to the higher signal detection rate on “SteamDrop” slides. Low chromatin accessibility may lead to weak or lack Tyr-FISH signals. The chromatin accessibility depends on chromosome compaction, presence of a layer of cytoplasm covering chromosomes, as well as on chromatin damage during the preparation of chromosome slides. These parameters may cause low frequency of probe hybridization and the signal-to-noise ratio. In order to increase the chromatin accessibility, less compact pachytene chromosomes [41], interphase nuclei [42], and DNA fiber [26] were used. Although fiber-FISH and FISH on interphase nuclei provide superior accessibility, they do not allow assigning signals to specific chromosomes. We assessed the ways in which the method of slide preparation can affect the structure of chromatin in terms of its accessibility. DIC microscopy clearly showed that the chromosomes prepared by the “SteamDrop” method retain their three-dimensional structure and, therefore, provide a better sample accessibility, in comparison to the squashed chromosomes, which are denser and may have a potential risk of chromatin damage.

Most studies on Tyr-FISH visualization of short single DNA sequences have been done on animal or human chromosomes [43,44,45,46,47]. Typically, slides of human and animal chromosomes are prepared by dropping a cell suspension in a fixative containing methanol (or ethanol) and acetic acid at a ratio of 3:1. However, plant chromosome preparation is prone to be conducted with the squashed method, which may serve as an explanation of the generally unsuccessful attempts to use the Tyr-FISH on plant chromosomes.

### 3.2. A Dual-Color Tyr-FISH for Integration of Recombination and Cytogenetic Maps and Genome Assembly

We applied the dual-color Tyr-FISH method to visualize a short target DNA sequence on high-compacted onion chromosomes (250 Mb µm^−1^, [21]). Thorough study of every step of the protocol allowed us to optimize the protocol and propose a more reliable method with a shorter runtime. Slide pretreatment was shortened and included only paraformaldehyde crosslinking to preserve chromatin structure and dehydration in ascending concentrations of ethanol. The pipeline of the exon-intron probe design was developed using database and bioinformatics tools. The method allowed detecting the transcript marker (1.6 Kb) on physical chromosomes in 51.6% of analyzed metaphases. Owing to the fact that the method’s high sensitivity is caused by the deposition of numerous fluorescent molecules, it may limit the resolution on compacted mitotic plant chromosomes. In this case, pachytene [48] and even extended pachytene chromosomes [49] can be used to resolve the Tyr-FISH probes. Despite the tremendous progress in genome sequencing, the assembly of the plant genome at the chromosome level remains a challenge. According to the NCBI data, nearly 200 edible plants’ genomes were assembled at the chromosome level [50]. There are around 6000 plant taxa considered as crops in various cultures, meaning that the vast number of crop genomes supporting the world’s plant-based food production are neither assembled nor even sequenced [51].

A relatively good genome assembly relies not only on the assembling the contigs into scaffolds but also on the placement and contiguity of these scaffolds on chromosomes in order to form the physical map. For instance, *Allium cepa* genome assembly (GCA_905187595.1) has a total length of ~15 Gbp which is close to the total genome size of *A. cepa* (~16 Gbp) though only a part of it is placed into the placed sequence. The vast majority (~12.7 Gbp) of genome assembly is in unplaced scaffolds. In our previous study, we have physically mapped the marker ACM082 from the genetic map of *A. cepa* on chromosome 4 [11]. This marker is not presented in placed sequences of assembly. Alignment the sequence of this marker against unplaced scaffolds of *A. cepa* assembly showed its presence in one of the unplaced scaffolds. This unplaced scaffold has a length of ~677 Kb and can be used to improve assembly.

Therefore, a reproducible method for visualizing short DNA sequences can serve as a convenient tool for validation of bioinformatics genome assembly using cytogenetic anchor points that provide reliable support for the integration of sequence data. Fifteen years ago, J. Jiang and B. Gill prophetically wrote: “In the foreseeable future, whole-genome sequencing will no longer be a hurdle for any plant species. However, this does not spell the end of FISH as a physical mapping tool” [26].

## 4. Materials and Methods

### 4.1. Plant Materials

*Allium cepa*. L., var. “Haltsedon” (2*n* = 2*x* = 16) were grown in pots in a greenhouse under controlled conditions: 14 h photoperiod (REFLUX lamp 400 watts; light intensity: 8000 lx) at a temperature of 22 °C.

### 4.2. Chromosome Preparations

In order to arrest the chromosomes at metaphase stage young roots were submerged in a saturated aqueous solution of α-bromonaphthalene (1:1000, *v*/*v*) overnight at 4 °C. The root tips were fixed in freshly prepared 3:1 (*v*/*v*) ethanol:acetic acid mixture for 1 h at RT and stored at −20 °C.

#### 4.2.1. “SteamDrop” Method

Chromosome preparations were made according to the “SteamDrop” protocol [30]. The fixed roots were washed in water for 30 min and then in citrate buffer (10 mM sodium citrate, 10 mM citric acid) pH 4.8 for 5 min. Dissected meristems (1–5 meristems) were transferred to 0.5 mL tubes with 20–30 μL of 0.1% enzyme mixture (1:1:1) pectolyase Y-23 (Kikkoman, Tokyo, Japan), Cellulase Onozuka R-10 (Yakult Co. Ltd., Tokyo, Japan), and Cytohelicase (Sigma-Aldrich Co. LLC, St. Louis, MO 63103 USA) for 120 min at 37 °C. After proper digestion of the cell wall in enzyme mixture (see video [52]) the tube was plugged into the ice. A total of 600 μL of ice-cold TE were added and gently mixed to wash the root sections, followed by centrifugation at 6000 rpm for 45 s. A supernatant was removed and 600 μL of 96% ethanol were added and mixed. Prior to slide preparation cell suspension can be stored at −20 °C for up to 6 months or it may be proceeded with immediately. The cell suspension was centrifuged at 6000 rpm for 30 s. The supernatant was discarded by inverting the tube. The pellet was suspended in 20–100 μL of 96% ethanol, depending on the cell concentration. A total of 4 μL of cell suspension was dropped onto a slide and by the time the surface became granule-like (10–15 s), 30 μL of ethanol/acetic acid fixative at a ratio of 3:1 were added. When the surface became granule-like (25–35 s), the slides were put upside down under the steam coming from a water bath at 55 °C (at the height 10–15 cm from the water surface of the water bath) for 3–5 s. Then 30 μL of ethanol and acetic acid fixative were dropped on the slide in a ratio of 1:1. After the surface became granule-like (25–35 s), the slides were put upside down under the steam for 1 sec only and immediately dried with an airflow (e.g., a tabletop fan).

#### 4.2.2. Squash Method

Mitotic metaphase chromosomes were prepared from young root tips. The fixed roots were washed in water for 30 min by 10 mM of citrate buffer (pH 4.8) for 5 min. The roots were transferred to a Petri dish with a diameter of 35 mm containing 0.5 mL of 0.1% enzyme mixture (see above) and incubated for 75 min at 37 °C. The macerated root tips were spread through dissection and squashed in a drop of 45% acetic acid. The permanent slides were prepared using liquid nitrogen. After removing the coverslip with a razor blade, the slides were rinsed briefly in 96% ethanol and air-dried for at least one hour.

### 4.3. Probe Preparation

Primers for the probe preparation (Table 1) were designed to obtain an exon-intron PCR-product. A total of 25 µL of PCR mixture contained 2.5 µL of 10x Taq Turbo buffer (Evrogen, 25 mM MgCl_2_, pH = 8.6), 0.2 mM of each dNTP, 0.2 mM of each primer, 2.5 U of Taq DNA polymerase (Evrogen, Moscow, Russia), and 100 ng of genomic DNA of Allium cepa L. var. Haltsedon. Amplification was performed using following PCR program: 5 min of initial denaturation at 95 °C, 35 cycles of 95 °C—30 s, 60 °C—30 s, 72 °C—150 s, and 5 min of final elongation at 72 °C. PCR-product was checked using electrophoresis in 1% agarose gel (0.5x TBE, 5 V/cm). PCR-product was precipitated using ethanol and 3M potassium acetate accordingly to Wallace et al. [53] with modifications. A total of 20 µL of PCR-product was mixed with 2 µL of 3M potassium acetate and 66 µL (3 volumes) of 96% ethanol (−20 °C) using vortex and incubated for 1 h at −20 °C. Afterwards, the mixture was centrifuged in a benchtop centrifuge (14,000 rpm, 30 min, RT). Supernatant was carefully discarded, replaced with 1 mL of 70% ethanol (−20 °C), mixed and centrifuged (14,000 rpm, 10 min, RT). Supernatant was gently removed, pellet was air-dried and resuspended in 10 µL of ddH_2_O. Concentration of precipitated PCR-product was measured using NanoDrop ND-1000 (Thermo Fisher Scientific Inc., Waltham, Ma 02451 USA). AT-cloning of PCR-product was performed in a pAL2-T vector (Evrogen, Moscow, Russia) and inoculated into *E. coli* strain XL1-Blue (Evrogen, Moscow, Russia) using electroporation according to the manufacturer’s protocol. Ten-times excess of PCR-product (insert) to vector was used in ligation reaction. Blue-white screening was used to select colonies containing plasmid with an insert of interest. Selected colonies were screened using PCR with insert-specific primers and inoculated into 5 mL of LB medium (10 g/L tryptone, 5 g/L yeast extract, 5 g/L NaCl) for overnight culture and follow up plasmid DNA isolation. Plasmid DNA was isolated using a GeneJET Plasmid Miniprep Kit (Thermo Fisher Scientific Inc., Waltham, Ma 02451 USA) according to the manufacturer’s protocol. The concentration of isolated plasmid DNA was measured using NanoDrop ND-1000 (Thermo Fisher Scientific Inc., Waltham, Ma 02451 USA). A total of 1 µL of isolated plasmid DNA was checked using electrophoresis in 1% agarose gel (0.5x TBE, 5 V/cm). The 5′ end and 3′ end of the insert was Sanger sequenced using standard M13 primer set. Plasmid DNA was labeled with digoxigenin-11-dUTP or biotin-16-dUTP using DIG or Biotin-Nick Translation Mix respectively (Roche, Mannheim, Germany). Determination of the fragment’s length of labeled probe was checked using electrophoresis according to the manufacturer’s protocol.

### 4.4. Dual-Color Sequential Tyr-FISH

Protocol of Dual-color sequential Tyr-FISH


**Pretreatment**


1Incubate the slides in 4% (*w*/*v*) paraformaldehyde in water at RT for 10 min (USE THE FUME HOOD!).2Wash slides in 2xSSC at RT three times for 5 min.3Dehydrate slides for 2 min each in 70, 90, and 100% ethanol and air dry.


**Hybridization**


4Prepare the hybridization mixtures (40 μL per slide) *20 μL formamide8 μL 50% dextransulphate4 μL 20x SSC1 μL 10% SDSx μL probe DNA (50 ng/slide)z μL probe DNA (50 ng/slide)y μL water5Denature hybridization mix at 75 °C for 5 min and plunge directly into the ice for 3 min.6Add the appropriate hybridization mix to each slide and cover with 24 × 25 coverslip.7Denature the slides at 75 °C for 5 min.8Place the slides in prewarmed humid chamber and incubate overnight at 37 °C.


**Stringency Washing**


Stringent of washing—82%

9Wash slides in 2xSSC at 42 °C twice for 5 min (with agitation).10Wash slides in 0.1xSSC at 55 °C twice for 7 min (without agitation).11Wash slides in 2xSSC at 42 °C for 3 min (with agitation).12Take Coplin jar out of water bath and leave to cool to RT for 20–25 min.13Wash slides in 4xSSC at RT for 3 min.14Wash slides in 2xSSC at RT for 2 min (with agitation).


**Detection**


15Wash slides in fresh TNT (0.1 M Tris-HCl, 0.15 M NaCl, pH 7.5, 0.05% Tween 20) buffer at RT for 5 min with agitation.16Block slides with 100 μL TNB buffer (1% blocking reagent in TN buffer) and place a coverslip (25 × 50 mm) to reduce evaporation. Incubate the slides in a humid chamber at RT for 15 min.17Drain off TNB buffer. Add 100 μL of Streptavidin-HRP (PerkinElmer, Waltham, MA 02451 USA) (1:500 diluted in TNB) to each slide and place a coverslip (25 × 50 mm) on top to reduce evaporation. Incubate the slides in a humid chamber at RT for 40 min.18Wash slides three times in fresh TNT at RT with agitation for 5 min.19Pipet 200 μL of the TSA PLUS Cy3 Reagent (Akoya Biosciences, Menlo Park, CA 94025 USA) (1:50 in 10% dextransulphate in 1x Amplification) onto each slide. Incubate the slides at RT for 3 to 10 min **. (per one slide: 4 μL stock Tyramide-CY3 + 40 μL 50% dextransulphate + 156 μL 1xAmplification Diluent).20Wash slides three times in fresh TNT at RT with agitation for 5 min.21Deactivate the remaining HRP activity by adding 100 μL 3% H_2_O_2_ in 1xTN (0.1 M Tris-HCl, pH 7.5, 0.15 M NaCl) for 30 min.22Wash the slides three times for 5 min each in TNT at RT with agitation.23Block slides with 100 μL 1% TNB buffer and place a coverslip (25 × 50 mm) to reduce evaporation. Incubate the slides in a humid chamber at RT for 15 min.24Drain off 1% TNB buffer. Add 100 μL of anti-digoxigenin HRP (Akoya Biosciences) (1:500 diluted in 1% TNB) to each slide and place a coverslip (25 × 50 mm) on top to reduce evaporation. Incubate the slides in a humid chamber at RT for 40 min.25Wash slides three times in fresh TNT at RT with agitation for 5 min.26Pipet 200 μL of the TSA PLUS Fluorescein Reagent (Akoya Biosciences) (1:50 in 10% dextransulphate in 1x Amplification) onto each slide. Incubate the slides at RT for 3 to 10 min (per one slide: 4 μL stock tyramide-FITC + 40 μL 50% dextransulphate + 156 μL 1xAmplification Diluent).27Wash slides three times in fresh TNT at RT with agitation for 5 min.


**Counterstaining**


28Wash slides in 2xSSC at RT three times for 5 min.29Dehydrate slides for 2 min each in 70, 90, and 100% ethanol and air dry in dark place.30Prepare fresh 1:20 dilution of DAPI in Vectashield. Add 20 µL per slide. Apply a 24 × 50 mm coverslip.


**Note**


*—mix all components of the hybridization mixture without adding DNA probes. Mix well by vortex until homogenous. Add probes in homogeneous mixture and mix briefly on vortex and centrifuge.

**—incubation time with tyramides depends on target size. For the target size of 13.3 Kb, 3 min is enough [21]. For the target size of 2–3 Kb, 7–10 min is enough (in this paper). If the incubation time was longer, the signal would appear as a strong band instead of two spots on both sister chromatids.

### 4.5. Microscopy

For differential interference contrast (DIC) microscopy the air-dried chromosome preparations obtained by both “SteamDrop” and squashed methods were directly examined under the Imager.D1 microscope, (http://www.zeiss.com, accessed on 16 April 2021), using an immersion objective x100 equipped with DIC.

For fluorescent microscopy the slides were examined under a Zeiss AxioImager M2 microscope (http://www.zeiss.com, accessed on 16 April 2021). The selected images were captured using a digital Hamamatsu camera C13440-20CU (http://www.hamamatsu.com accessed on 16 April 2021). Image processing was performed by Zen 2.6 (blue edition) an image analysis software. The captured images of the chromosomes and position of Tyr-FISH signals were measured using the program DRAWID [36]. Only non-overlapping chromosomes were used for the measurement of the positions of Tyr-FISH signals. The relative position of hybridization sites on chromosomes (RPHC) was calculated in the form of the ratio of the distance between the site of hybridization and the centromere to the length of the chromosome arm. Karyotype analysis was performed according to the standard onion nomenclature system proposed by Kalkman [54] and confirmed by the Fouth Eucarpia *Allium* Symposium [55].

## Figures and Tables

**Figure 1 ijms-22-05860-f001:**
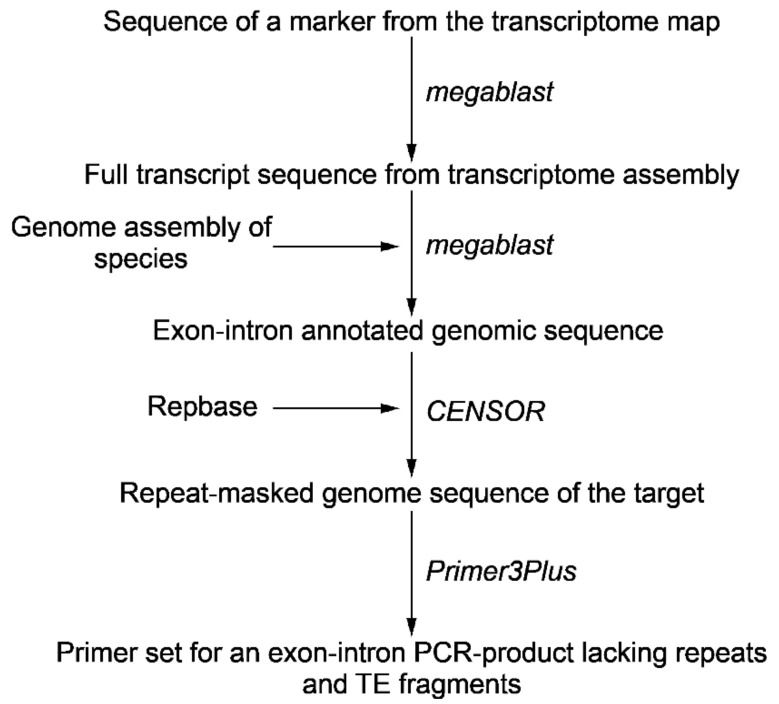
Pipeline in order to design an exon-intron probe for Tyr-FISH.

**Figure 2 ijms-22-05860-f002:**
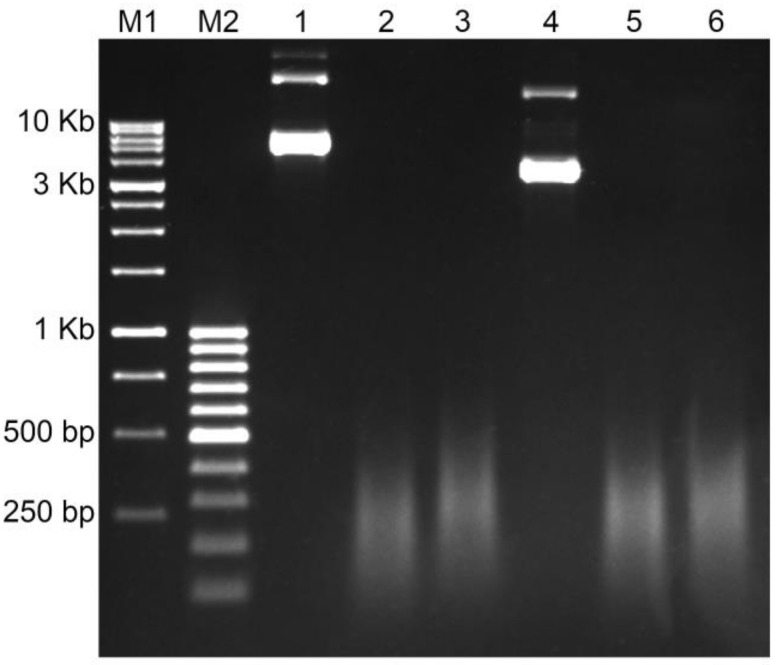
The results of the Nick translation of the plasmid DNA containing insert of the genomic amplicon in 1% agarose gel: M1—DNA Ladder 1 Kb; M2—DNA Ladder 100 bp+; lines 1–3—Unigene572 (insert size is 3.2 Kb): line 1—the non-labeled plasmid DNA, line 2—labeling with Dig-11-dUTP, line 3—labeling with Biotin-16-dUTP; lines 4–6—Unigene5305 (insert size is 1.6 Kb): line 4—the non-labeled plasmid DNA, line 5—labeling with Dig-11-dUTP, line 6—labeling with Biotin-16-dUTP.

**Figure 3 ijms-22-05860-f003:**
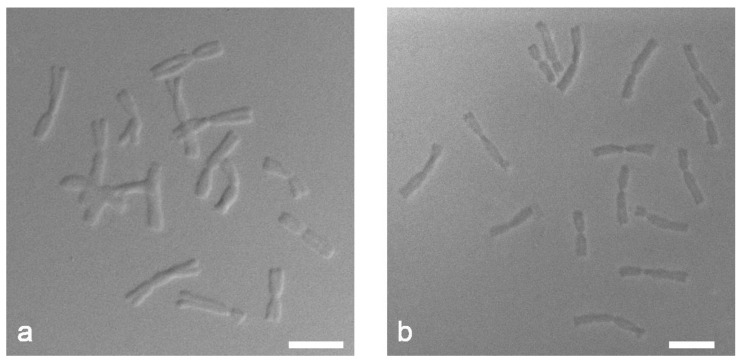
Differential interference contrast (DIC) microscopy of mitotic metaphase chromosomes of *Allium cepa*: (**a**) “SteamDrop” method of chromosome slide preparation; (**b**) squashed method of chromosome slide preparation. Scale bar—10 µm.

**Figure 4 ijms-22-05860-f004:**
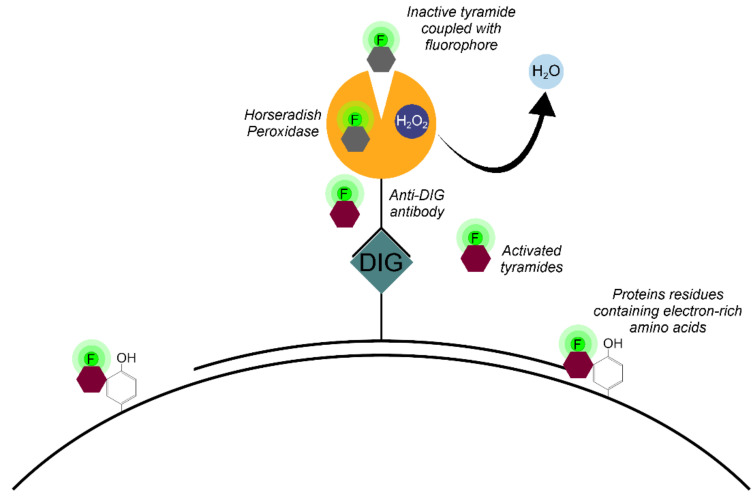
Principle of signal detection on physical chromosomes using horseradish peroxidase (HRP) and fluorophore-labeled tyramides.

**Figure 5 ijms-22-05860-f005:**
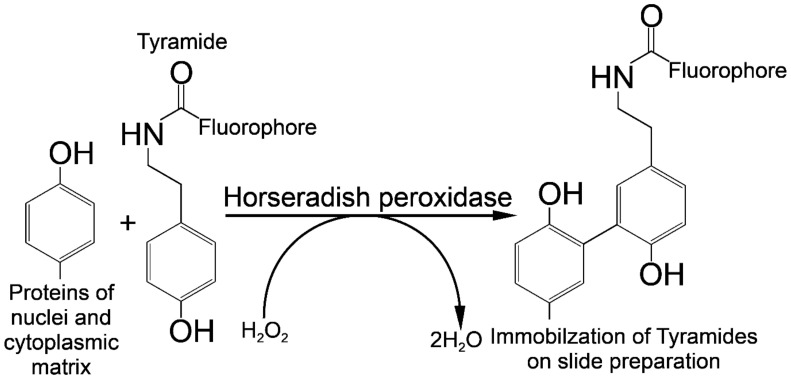
Horseradish peroxidase is a hydrogen peroxide (H_2_O_2_) decomposing enzyme concomitant with the oxidation of phenolic substrate (tyramide) and reaction with electron-rich moieties of proteins of nuclear and cytoplasmic matrixes.

**Figure 6 ijms-22-05860-f006:**
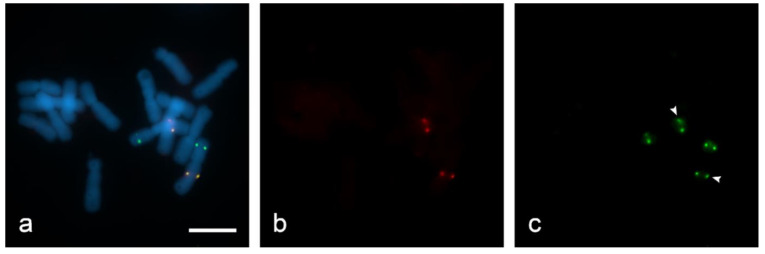
Evaluation of quenching of horseradish peroxidase (HRP) in dual-color sequential Tyr-FISH “SteamDrop” slide. Quenching with 0.3% H_2_O_2_ for 30 min: (**a**) visualization of two probes—merged image; (**b**) visualization of Unigene572 detected with tyramide-Cy3 using Cy3-filter (red); (**c**) visualization of Unigene5305 detected with Tyr-FITC using FITC-filter (green), arrows indicate signals arising from Unigene572 because HRP activity after the first detection step was not quenched. Scale bar—10 µm.

**Figure 7 ijms-22-05860-f007:**
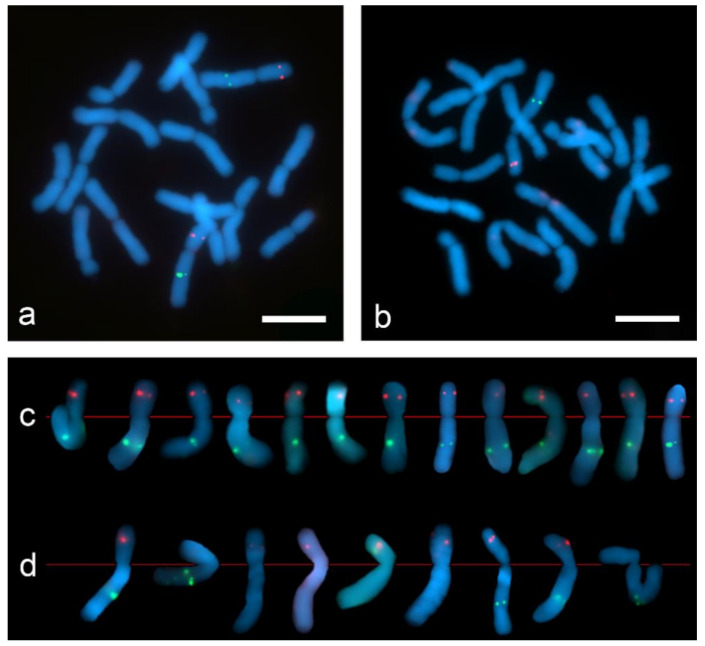
Dual-color Tyr-FISH on mitotic metaphase chromosomes of *Allium cepa* probing with Unigene572 (red) and Unigene5305 (green): (**a**) “SteamDrop” chromosome preparation; (**b**) squashed chromosome preparation; (**c**) extracted chromosome 2 from metaphases of “SteamDrop” chromosome preparations; (**d**) extracted chromosome 2 from metaphases of squashed chromosome preparation. Scale bar—10 µm.

**Figure 8 ijms-22-05860-f008:**
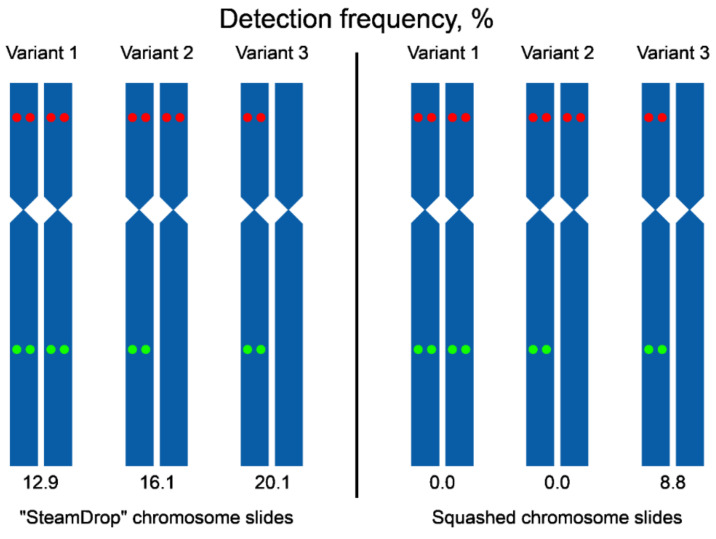
Scheme of variants of a signal detection on a pair of homologous chromosomes 2 using chromosome preparations obtained by “SteamDrop” and squashed methods. The green dots –Unigene5305, the red dots – Unigene572.

**Figure 9 ijms-22-05860-f009:**
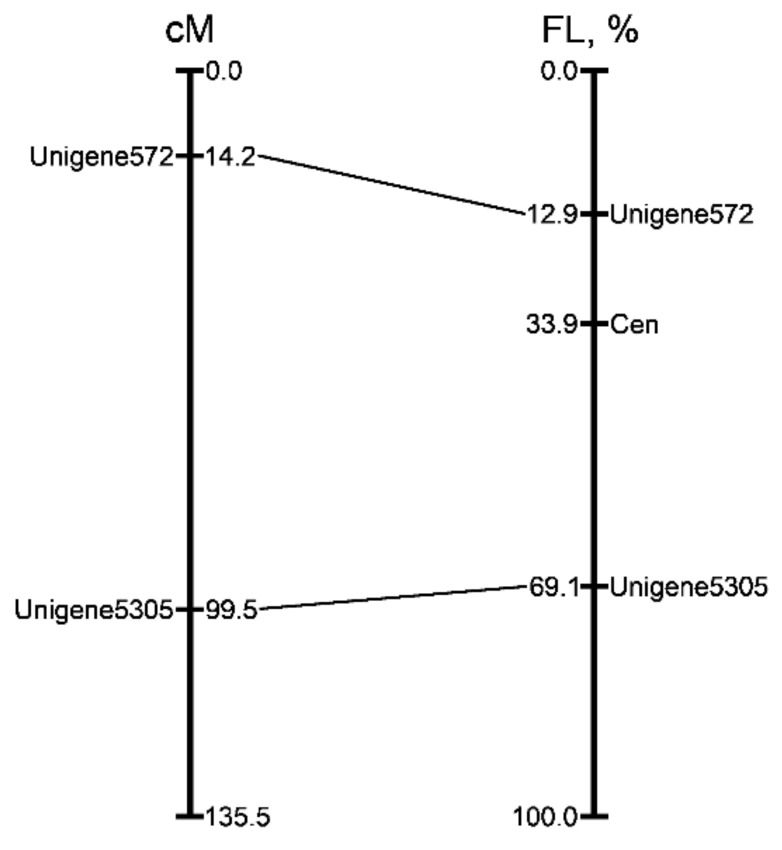
Alignment of the genetic and cytogenetic maps of onion chromosome 2. The position of the markers (figure on the left) in genetic map described by Fujito et al. [27]. Distances in centimorgans are shown on the right of linkage group. The physical positions (figure on the right) of two mapped markers are expressed as a percentage of the fractional length (distance from the end of the short arm to the signals divided by the length of the entire chromosome). Corresponding positions on the genetic map are indicated with lines.

**Table 1 ijms-22-05860-t001:** Sequences of primers used to produce genomic amplicons for Tyr-FISH.

Probe Name	Direction	Primer Sequence (5′ to 3′)	Size
Unigene572	F	AGTGGTGCAGTTCTTCAGCA	3.2 Kb
	R	AACCGATTGGCAGGGAAGTT	
Unigene5305	F	TGTTTCACAAACGCTTCGTCC	1.6 Kb
	R	TGTCGCCACCACTCATTCAA	

**Table 2 ijms-22-05860-t002:** Influence of the method of chromosome slide preparation on the frequency of signal detection.

Method	Probe	Number of Analyzed Metaphases	Number of Metaphases with Signal			Signal Detection Frequency, %		
			Total	On One Homolog	On Both Homologs	Total MetaPhases with Signal	On One Homolog	On Both Homologs
Squashed	Unigene572 (3.2 Kb)	34	12	11	1	35.3	32.4	2.9
	Unigene5305 (1.6 Kb)		5	5	0	14.7	14.7	0.0
SteamDrop	Unigene572 (3.2 Kb)	62	48	22	26	77.4	35.5	41.9
	Unigene5305 (1.6 Kb)		32	24	8	51.6	38.7	12.9

## Data Availability

Not applicable.

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
