# Peer review of "A Dual-Color Tyr-FISH Method for Visualizing Genes/Markers on Plant Chromosomes to Create Integrated Genetic and Cytogenetic Maps"

_ijms, 2021, doi:10.3390/ijms22115860_

Round 1

Reviewer 1 Report

General comment:

The authors introduced protocol for an ultrasensitive dual-color Tyr-FISH for two short unique DNA sequences (Unigene 572 and 5305) on chromosome 2 of Allium cepa. L.  Many factors such as chromosome spreading method, probe design and labelling, stringency washing were investigated in order to establish the optimal protocol for detecting short DNA sequence on plant chromosome.

The result demonstrated that the metaphase spread method was extremely important step in Tyr-FISH on plant chromosome. In addition, the pipeline to design probe as well as labelling steps have been detailed.

Details comments and suggestions:

  • While Tyr-FISH with individual single copy probes could be useful to uncover order and physical distance on a chromosome, for supporting chromosome-scale genome assembly (as the authors claim for Tyr-FISH), Hi-C and oligo-FISH are much easier approaches. Still the probe generation for Tyr-FISH is very laborious and the efficiency (which was not compared to that of conventional FISH with directly labeled probes) is still not very high. What is actually novel compared to their previous publications remains unclear.
  • Are the sequences of exon-intron PCR-product available and accessible on database?
  • It could be more reader friendly when all steps of Tyr-FISH are shown in the diagram
  • Could the “Steam Drop” slides be reused for further Tyr-FISH round in order to integrated several markers on the same chromosomes?
  • On p. 2, 69-70, the cytogenetic approach detected interchromosomal rearrangements rather than inversion
  • It should consistently read drop/squash method (a method cannot be squashed or dropped).
  • 3, 117-118 should read (M. Shiguyo unpublished).
  • Legend to Fig. 2 should read: …lines 1-3….
  • 5, 159-162: The meaning of this sentence is unclear. Lines 167-168 to mention nationality and profession of an author is unusual.
  • In Fig. 4 it is not mentioned how HRR interacts with the labeled probe.
  • 7, 207 Of course, other FISH methods can easily detect more probes simultaneously!
  • 12, 393: (~12.7Gbp) instead of (~12.7Gpb)

Conclusion:

              The paper needs a careful check of English language regarding style and grammar. It could be accepted after revision according recommendations of reviewers and editors.

Author Response

We thank the Reviewer for a positive assessment of our research and useful comments and suggestions.

Please find our answer to comments (Your comments are indicated by black dots)

  • While Tyr-FISH with individual single copy probes could be useful to uncover order and physical distance on a chromosome, for supporting chromosome-scale genome assembly (as the authors claim for Tyr-FISH), Hi-C and oligo-FISH are much easier approaches. Still the probe generation for Tyr-FISH is very laborious

The method we propose is not more complicated than other methods. Signal detection with Tyramids is much faster than conventional FISH.  Making a labeled DNA probe is no more difficult than for GISH or FISH and much easier than the laborious BAC library creation. During the three weeks that our article was uploaded to the preprint server, there were 46 downloads of the article.

  • What is actually novel compared to their previous publications remains unclear

Compared to our previous publications all steps of the protocol leading to the successful visualization of short DNA sequences have been studied in detail. Critical milestones were identified and optimized. The principles of the method at the atomic-molecular level were explained, starting with the preparation of chromosomes and ending with the detection of the signal, which was not described so extensively before. Only fragmented data on the work of peroxidase in articles on physiology and biochemistry or its use in immunochemistry were reported. We hope that our experience will help other researchers actively use this method in their work.

  • Are the sequences of exon-intron PCR-product available and accessible on database?

NCBI requests annotation of sequences. For visualization of markers, genomic amplicons, which are only a part of genes, were sufficient. Complete gene sequencing was not the goal of these studies.

The amplicon sequences of the Tyr-FISH probes (Unigene572 and Unigene5305) attached in the Supplementary file.

  • Could the “Steam Drop” slides be reused for further Tyr-FISH round in order to integrated several markers on the same chromosomes?

Yes, the “Steam Drop” slides could be reused for further Tyr-FISH round in order to integrated several markers on the same chromosomes.  Now commercially available only fluorochrome conjugated tyramides with green and red emission spectra.

  • On p. 2, 69-70, the cytogenetic approach detected interchromosomal rearrangements rather than inversion

Here we are talking about inverted orientation of a sequence within a scaffold or pseudomolecule, but not about cytological detection of chromosome inversion. Such conclusions are made by the authors of the article, to which we refer (Hoang, P.N.T.; Michael, T.P.; Gilbert, S.; Chu, P.; S Motley, T.; Appenroth, K. J.; Schubert, I.; Lam, E. Generating a high-confidence reference genome map of the Greater Duckweed by integration of cytogenomic, optical mapping and Oxford Nanopore technologies. Plant J. 2018, 96, 670–684. doi: 10.1111/tpj.14049)

Earlier, we showed a discrepancy between the genome assembly and the order of BAC clones on the physical chromosome in tomato (Szinay, D.; Chang, S.-B.; Khrustaleva, L.; Peters, S.; Schijlen E.; Bai, Y.; Stiekema, W. J.; van Ham, R. C. H. J.; de Jong, H.; Lankhorst, M. R. K.; High-resolution chromosome mapping of BACs using multi-colour FISH and pooled-BAC FISH as a backbone for sequencing tomato chromosome 6. Plant J 2008, 56, 627–637. doi: 10.1111/j.1365-313X.2008.03626.x.)

  • It should consistently read drop/squash method (a method cannot be squashed or dropped)

These two methods are different: The SteamDrop method is based on using a suspension of cells and dropping cells onto glass without squashing, while the squashed method uses squashing cells on glass.

  • 3, 117-118 should read (M. Shiguyo unpublished).

 (the draft genome assembly of A. cepa received from Dr. Shigyo) replaced by (S. Sato & M. Shigyo unpublished)

  • Legend to Fig. 2 should read: …lines 1-3….

Thank you. Replaced by “lines 1-3”

  • 5, 159-162: The meaning of this sentence is unclear.

Thank you. Replaced by “the non-labeled plasmid DNA, line 5 - labelling with Dig-11-dUTP, line 6 - labelling with Bio-tin-16-dUTP”.

  • Lines 167-168 to mention nationality and profession of an author is unusual.

We replaced it with a more concise sentence: “The technique was developed by Georges Nomarski in 1952”.

  • In Fig. 4 it is not mentioned how HRR interacts with the labeled probe.

 We have made changes to the picture.

HRP is conjugated with anti-digoxigenin, which is antibody to digoxigenin.  With anti-digoxigenin-HRP was detected digoxigenin-labeled probe.

  • 12, 393: (~12.7Gbp) instead of (~12.7Gpb)

Thank you, corrected to (~12.7Gbp)

  • The paper needs a careful check of English language regarding style and grammar.

The text has been carefully edited with the help of a translator.

Reviewer 2 Report

The physical mapping of marker sequences is a very important tool aiding the generation of genetic maps and genome assembly at chromosomal level. However, physical mapping of short sequences that are unique or low repeated sequences is still challenging.

This paper addresses the potential for the application of the Tyr FISH technique (a sensitive techniques for in situ mapping of genes onto the chromosomes) to plant chromosomes. The methodological approach is appropriate, clearly described and supported. Figures are self explanatory and well conceived. Discussion is performed in a rather limited context, but it is well focused, and keeps the paper concise and to the point. The protocol developed could greatly enhance the accuracy in determining the relative position of genetic markers in plant genomes, thus contributing to improved integration of genetic and cytogenetic mapping.

The use of DIC to detect and describe differences in the quality of chromosome morphology from different preparations is interesting, and it is worth noting that the differences detected through DIC microscopy (not evident after DAPI staining, as far as one can tell by looking at the images presented in the paper) are reflected in the marked differences of the Tyr-FISH results. 

Overall, I enjoyed reading this paper, and, even if it is mainly a methodology paper, I strongly support publication.

A few minor comments:

Lines 363 – 364 “Most studies on tyr-FISH visualization of short single DNA sequences have been done on animal or human chromosomes” are mainly mammalian species or other vertebrates/invertebrates have been tested? Just curious. 

Lines 482 – 483 was the 4% paraformaldehyde in PBS or 2XSSC? Please specify. 

Author Response

We thank the Reviewer for their positive comment and careful review

Please find our answer to comments (Your comments are indicated by arrows)

  • Lines 363 – 364 “Most studies on tyr-FISH visualization of short single DNA sequences have been done on animal or human chromosomes” are mainly mammalian species or other vertebrates/invertebrates have been tested? Just curious. 

Tyr-FISH studies have been done mainly on mammalian species. As far as we know the method was used on Western clawed frog (Xenopus tropicalis) (Krylov et al., 2007) and grasshopper (Navarro-Domínguez B. et al., 2017).

  • Lines 482 – 483 was the 4% paraformaldehyde in PBS or 2XSSC?

Added: in 4% (w/v) paraformaldehyde in water 

Round 2

Reviewer 1 Report

  • While Tyr-FISH with individual single copy probes could be useful to uncover order and physical distance on a chromosome, for supporting chromosome-scale genome assembly (as the authors claim for Tyr-FISH), Hi-C and oligo-FISH are much easier approaches. Still the probe generation for Tyr-FISH is very laborious

The method we propose is not more complicated than other methods. Signal detection with Tyramids is much faster than conventional FISH.  Making a labeled DNA probe is no more difficult than for GISH or FISH and much easier than the laborious BAC library creation. During the three weeks that our article was uploaded to the preprint server, there were 46 downloads of the article.

I mean Hi-C and oligo-FISH are much easier approaches than Tyr-FISH for supporting chromosome-scale genome assembly

  • What is actually novel compared to their previous publications remains unclear

Compared to our previous publications all steps of the protocol leading to the successful visualization of short DNA sequences have been studied in detail. Critical milestones were identified and optimized. The principles of the method at the atomic-molecular level were explained, starting with the preparation of chromosomes and ending with the detection of the signal, which was not described so extensively before. Only fragmented data on the work of peroxidase in articles on physiology and biochemistry or its use in immunochemistry were reported. We hope that our experience will help other researchers actively use this method in their work.

             Fine

  • Are the sequences of exon-intron PCR-product available and accessible on database?

NCBI requests annotation of sequences. For visualization of markers, genomic amplicons, which are only a part of genes, were sufficient. Complete gene sequencing was not the goal of these studies.

The amplicon sequences of the Tyr-FISH probes (Unigene572 and Unigene5305) attached in the Supplementary file.

Great

  • Could the “Steam Drop” slides be reused for further Tyr-FISH round in order to integrated several markers on the same chromosomes?

Yes, the “Steam Drop” slides could be reused for further Tyr-FISH round in order to integrated several markers on the same chromosomes.  Now commercially available only fluorochrome conjugated tyramides with green and red emission spectra.

Great

  • On p. 2, 69-70, the cytogenetic approach detected interchromosomal rearrangements rather than inversion

Here we are talking about inverted orientation of a sequence within a scaffold or pseudomolecule, but not about cytological detection of chromosome inversion. Such conclusions are made by the authors of the article, to which we refer (Hoang, P.N.T.; Michael, T.P.; Gilbert, S.; Chu, P.; S Motley, T.; Appenroth, K. J.; Schubert, I.; Lam, E. Generating a high-confidence reference genome map of the Greater Duckweed by integration of cytogenomic, optical mapping and Oxford Nanopore technologies. Plant J. 2018, 96, 670–684. doi: 10.1111/tpj.14049)

          Fine

Earlier, we showed a discrepancy between the genome assembly and the order of BAC clones on the physical chromosome in tomato (Szinay, D.; Chang, S.-B.; Khrustaleva, L.; Peters, S.; Schijlen E.; Bai, Y.; Stiekema, W. J.; van Ham, R. C. H. J.; de Jong, H.; Lankhorst, M. R. K.; High-resolution chromosome mapping of BACs using multi-colour FISH and pooled-BAC FISH as a backbone for sequencing tomato chromosome 6. Plant J 2008, 56, 627–637. doi: 10.1111/j.1365-313X.2008.03626.x.)

              Great

  • It should consistently read drop/squash method (a method cannot be squashed or dropped)

These two methods are different: The SteamDrop method is based on using a suspension of cells and dropping cells onto glass without squashing, while the squashed method uses squashing cells on glass.

Drop and squash methods are totally different, of course. I mean that it should read drop method (not dropped method) and squash method (not squashed method). Because the methods can not be dropped or squashed, only the samples are dropped or squashed

  • 3, 117-118 should read (M. Shiguyo unpublished).

 (the draft genome assembly of A. cepa received from Dr. Shigyo) replaced by (S. Sato & M. Shigyo unpublished)

 Fine

  • 5, 159-162: The meaning of this sentence is unclear.

Thank you. Replaced by “the non-labeled plasmid DNA, line 5 - labelling with Dig-11-dUTP, line 6 - labelling with Bio-tin-16-dUTP”.

Great

  • Lines 167-168 to mention nationality and profession of an author is unusual.

We replaced it with a more concise sentence: “The technique was developed by Georges Nomarski in 1952”.

Great

  • In Fig. 4 it is not mentioned how HRR interacts with the labeled probe.

 We have made changes to the picture.

Great

HRP is conjugated with anti-digoxigenin, which is antibody to digoxigenin.  With anti-digoxigenin-HRP was detected digoxigenin-labeled probe.

  • The paper needs a careful check of English language regarding style and grammar.

The text has been carefully edited with the help of a translator.

Great, this version is better.